# Personality types revisited–a literature-informed and data-driven approach to an integration of prototypical and dimensional constructs of personality description

**André Kerber**[1]*, **Marcus Roth**[2], **Philipp Yorck Herzberg**[3]

**1** Department of Psychology, Freie Universität Berlin, Berlin, Germany, **2** Department of Psychology, University of Duisburg-Essen, Duisburg Germany, **3** Personality Psychology and Psychological Assessment Unit, Helmut Schmidt University of the Federal Armed Forces Hamburg, Hamburg, Germany

\* andre.kerber@fu-berlin.de

## Abstract

A new algorithmic approach to personality prototyping based on Big Five traits was applied to a large representative and longitudinal German dataset (N = 22,820) including behavior, personality and health correlates. We applied three different clustering techniques, latent profile analysis, the k-means method and spectral clustering algorithms. The resulting cluster centers, i.e. the personality prototypes, were evaluated using a large number of internal and external validity criteria including health, locus of control, self-esteem, impulsivity, risk-taking and wellbeing. The best-fitting prototypical personality profiles were labeled according to their Euclidean distances to averaged personality type profiles identified in a review of previous studies on personality types. This procedure yielded a five-cluster solution: resilient, overcontroller, undercontroller, reserved and vulnerable-resilient. Reliability and construct validity could be confirmed. We discuss wether personality types could comprise a bridge between personality and clinical psychology as well as between developmental psychology and resilience research.

## Introduction

Although documented theories about personality types reach back more than 2000 years (i.e. Hippocrates' humoral pathology), and stereotypes for describing human personality are also widely used in everyday psychology, the descriptive and variable-oriented assessment of personality, i.e. the description of personality on five or six trait domains, has nowadays consolidated its position in modern personality psychology.

In recent years, however, the person-oriented approach, i.e. the description of an individual personality by its similarity to frequently occurring prototypical expressions, has amended the variable-oriented approach with the addition of valuable insights into the description of personality and the prediction of behavior. Focusing on the trait configurations, the person-oriented approach aims to identify personality types that share the same typical personality profile [1].

**Data Availability Statement:** The data used in this article were made available by the German Socio-Economic Panel (SOEP, Data for years 1984-2015)

at the German Institute for Economic Research, Berlin, Germany. To ensure the confidentiality of respondents' information, the SOEP adheres to strict security standards in the provision of SOEP data. The data are reserved exclusively for research use, that is, they are provided only to the scientific community. To require full access to the data used in this study, it is required to sign a data distribution contract. All contact informations and the procedure to request the data can be obtained at: https://www.diw.de/en/diw_02.c.222829.en/access_and_ordering.html.

**Funding:** The author(s) received no specific funding for this work.

**Competing interests:** The authors have declared that no competing interests exist.

Nevertheless, the direct comparison of the utility of person-oriented vs. variable-oriented approaches to personality description yielded mixed results. For example Costa, Herbst, McCrae, Samuels and Ozer [2] found a higher amount of explained variance in predicting global functioning, geriatric depression or personality disorders for the variable-centered approach using Big Five personality dimensions. But these results also reflect a methodological caveat of this approach, as the categorical simplification of dimensionally assessed variables logically explains less variance. Despite this, the person-centered approach was found to heighten the predictability of a person's behavior [3, 4] or the development of adolescents in terms of internalizing and externalizing symptoms or academic success [5, 6], problem behavior, delinquency and depression [7] or anxiety symptoms [8], as well as stress responses [9] and social attitudes [10]. It has also led to new insights into the function of personality in the context of other constructs such as adjustment [2], coping behavior [11], behavioral activation and inhibition [12], subjective and objective health [13] or political orientation [14], and has greater predictive power in explaining longitudinally measured individual differences in more temperamental outcomes such as aggressiveness [15].

However, there is an ongoing debate about the appropriate number and characteristics of personality prototypes and whether they perhaps constitute an methodological artifact [16].

With the present paper, we would like to make a substantial contribution to this debate. In the following, we first provide a short review of the personality type literature to identify personality types that were frequently replicated and calculate averaged prototypical profiles based on these previous findings. We then apply multiple clustering algorithms on a large German dataset and use those prototypical profiles generated in the first step to match the results of our cluster analysis to previously found personality types by their Euclidean distance in the 5-dimensional space defined by the Big Five traits. This procedure allows us to reliably link the personality prototypes found in our study to previous empirical evidence, an important analysis step lacking in most previous studies on this topic.

## The empirical ground of personality types

The early studies applying modern psychological statistics to investigate personality types worked with the Q-sort procedure [1, 15, 17], and differed in the number of Q-factors. With the Q-Sort method, statements about a target person must be brought in an order depending on how characteristic they are for this person. Based on this Q-Sort data, prototypes can be generated using Q-Factor Analysis, also called inverse factor analysis. As inverse factor analysis is basically interchanging variables and persons in the data matrix, the resulting factors of a Q-factor analysis are prototypical personality profiles and not hypothetical or latent variable dimensions. On this basis, personality types (groups of people with similar personalities) can be formed in a second step by assigning each person to the prototype with whose profile his or her profile correlates most closely. All of these early studies determined at least three prototypes, which were labeled resilient, overcontroler and undercontroler grounded in Block's theory of ego-control and ego-resiliency [18]. According to Jack and Jeanne Block's decade long research, individuals high in ego-control (i.e. the overcontroler type) tend to appear constrained and inhibited in their actions and emotional expressivity. They may have difficulty making decisions and thus be non-impulsive or unnecessarily deny themselves pleasure or gratification. Children classified with this type in the studies by Block tend towards internalizing behavior. Individuals low in ego-control (i.e. the undercontroler type), on the other hand, are characterized by higher expressivity, a limited ability to delay gratification, being relatively unattached to social standards or customs, and having a higher propensity to risky behavior. Children classified with this type in the studies by Block tend towards externalizing behavior.

Individuals high in Ego-resiliency (i.e. the resilient type) are postulated to be able to resourcefully adapt to changing situations and circumstances, to tend to show a diverse repertoire of behavioral reactions and to be able to have a good and objective representation of the "goodness of fit" of their behavior to the situations/people they encounter. This good adjustment may result in high levels of self-confidence and a higher possibility to experience positive affect.

Another widely used approach to find prototypes within a dataset is cluster analysis. In the field of personality type research, one of the first studies based on this method was conducted by Caspi and Silva [19], who applied the SPSS Quick Cluster algorithm to behavioral ratings of 3-year-olds, yielding five prototypes: undercontrolled, inhibited, confident, reserved, and well-adjusted.

While the inhibited type was quite similar to Block's overcontrolled type [18] and the well-adjusted type was very similar to the resilient type, two further prototypes were added: confident and reserved. The confident type was described as easy and responsive in social interaction, eager to do exercises and as having no or few problems to be separated from the parents. The reserved type showed shyness and discomfort in test situations but without decreased reaction speed compared to the inhibited type. In a follow-up measurement as part of the Dunedin Study in 2003 [20], the children who were classified into one of the five types at age 3 were administered the MPQ at age 26, including the assessment of their individual Big Five profile. Well-adjusteds and confidents had almost the same profiles (below-average neuroticism and above average on all other scales except for extraversion, which was higher for the confident type); undercontrollers had low levels of openness, conscientiousness and openness to experience; reserveds and inhibiteds had below-average extraversion and openness to experience, whereas inhibiteds additionally had high levels of conscientiousness and above-average neuroticism.

Following these studies, a series of studies based on cluster analysis, using the Ward's followed by K-means algorithm, according to Blashfield & Aldenderfer [21], on Big Five data were published. The majority of the studies examining samples with N < 1000 [5, 7, 22–26] found that three-cluster solutions, namely resilients, overcontrollers and undercontrollers, fitted the data the best. Based on internal and external fit indices, Barbaranelli [27] found that a three-cluster and a four-cluster solution were equally suitable, while Gramzow [28] found a four-cluster solution with the addition of the reserved type already published by Caspi et al. [19, 20]. Roth and Collani [10] found that a five-cluster solution fitted the data the best. Using the method of latent profile analysis, Merz and Roesch [29] found a 3-cluster, Favini et al. [6] found a 4-cluster solution and Kinnunen et al. [13] found a 5-cluster solution to be most appropriate.

Studies examining larger samples of N > 1000 reveal a different picture. Several favor a five-cluster solution [30–34] while others favor three clusters [8, 35]. Specht et al. [36] examined large German and Australian samples and found a three-cluster solution to be suitable for the German sample and a four-cluster solution to be suitable for the Australian sample. Four cluster solutions were also found to be most suitable to Australian [37] and Chinese [38] samples. In a recent publication, the authors cluster-analysed very large datasets on Big Five personality comprising more than 1,5 million online participants using Gaussian mixture models [39]. Albeit their results "provide compelling evidence, both quantitatively and qualitatively, for at least four distinct personality types", two of the four personality types in their study had trait profiles not found previously and all four types were given labels unrelated to previous findings and theory. Another recent publication [40] cluster-analysing data of over 270,000 participants on HEXACO personality "provided evidence that a five-profile solution was optimal". Despite limitations concerning the comparability of HEXACO trait profiles with FFM

personality type profiles, the authors again decided to label their personality types unrelated to previous findings instead using agency-communion and attachment theories.

We did not include studies in this literature review, which had fewer than 199 participants or those which restricted the number of types a priori and did not use any method to compare different clustering solutions. We have made these decisions because a too low sample size increases the probability of the clustering results being artefacts. Further, a priori limitation of the clustering results to a certain number of personality types is not well reasonable on the base of previous empirical evidence and again may produce artefacts, if the a priori assumed number of clusters does not fit the data well.

To gain a better overview, we extracted all available z-scores from all samples of the above-described studies. Fig 1 shows the averaged z-scores extracted from the results of FFM clustering solutions for all personality prototypes that occurred in more than one study. The error bars represent the standard deviation of the distribution of the z-scores of the respective trait within the same personality type throughout the different studies. Taken together the resilient type was replicated in all 19 of the mentioned studies, the overcontroler type in 16, the undercontroler personality type in 17 studies, the reserved personality type was replicated in 6 different studies, the confident personality type in 4 and the non-desirable type was replicated twice.

Three implications can be drawn from this figure. First, although the results of 19 studies on 26 samples with a total N of 1,560,418 were aggregated, the Big Five profiles for all types

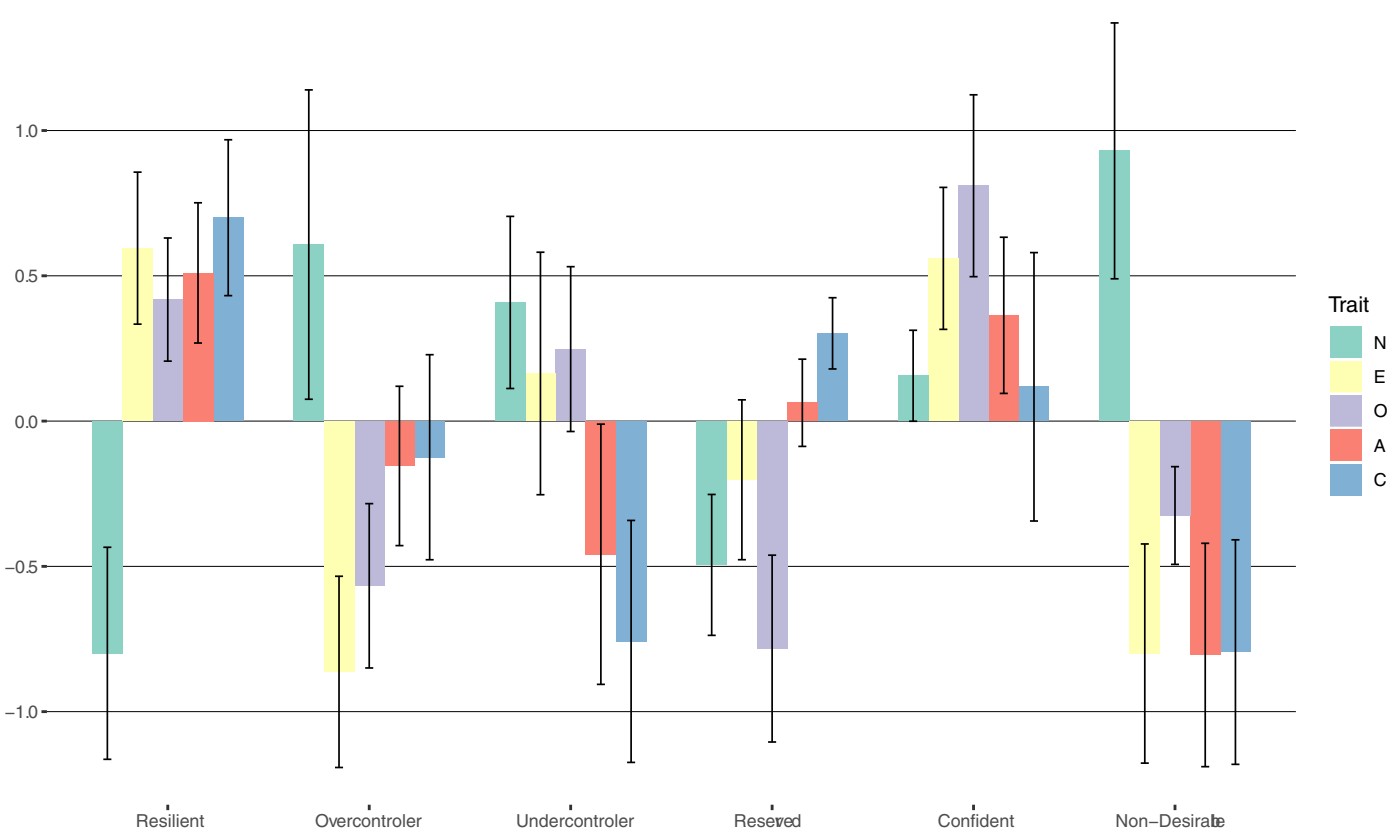

**Fig 1. Average Big Five z-scores of personality types found in previous literature.** Average Big Five z-scores of personality types based on clustering of FFM datasets with N ≥ 199 that were replicated at least once. Error bars indicate the standard deviation of the repective trait within the respective personality type found in the literature [5, 6, 10, 22–25, 27–31, 33–36, 38, 39, 41].

can still be clearly distinguished. In other words, personality types seem to be a phenomenon that survives the aggregation of data from different sources. Second, there are more than three replicable personality types, as there are other replicated personality types that seem to have a distinct Big Five profile, at least regarding the reserved and confident personality types. Third and lastly, the non-desirable type seems to constitute the opposite of the resilient type. Looking at two-cluster solutions on Big Five data personality types in the above-mentioned literature yields the resilient opposed to the non-desirable type. This and the fact that it was only replicated twice in the above mentioned studies points to the notion that it seems not to be a distinct type but rather a combined cluster of the over- and undercontroller personality types. Further, both studies with this type in the results did not find either the undercontroller or the overcontroller cluster or both. Taken together, five distinct personality types were consistently replicated in the literature, namely resilient, overcontroller, undercontroller, reserved and confident. However, inferring from the partly large error margin for some traits within some prototypes, not all personality traits seem to contribute evenly to the occurrence of the different prototypes. While for the overcontroler type, above average neuroticism, below average extraversion and openness seem to be distinctive, only below average conscientiousness and agreeableness seemed to be most characteristic for the undercontroler type. The reserved prototype was mostly characterized by below average openness and neuroticism with above average conscientiousness. Above average extraversion, openness and agreeableness seemed to be most distinctive for the confident type. Only for the resilient type, distinct expressions of all Big Five traits seemed to be equally significant, more precisely below average neuroticism and above average extraversion, openness, agreeableness and conscientiousness.

## Research gap and novelty of this study

The cluster methods used in most of the mentioned papers were the Ward's followed by K-means method or latent profile analysis. With the exception of Herzberg and Roth [30], Herzberg [33], Barbaranelli [27] and Steca et. al. [25], none of the studies used internal or external validity indices other than those which their respective algorithm (in most cases the SPSS software package) had already included. Gerlach et al. [39] used Gaussian mixture models in combination with density measures and likelihood measures.

The bias towards a smaller amount of clusters resulting from the utilization of just one replication index, e.g. Cohen's Kappa calculated by split-half cross-validation, which was ascertained by Breckenridge [42] and Overall & Magee [43], is probably the reason why a three-cluster solution is preferred in most studies. Herzberg and Roth [30] pointed to the study by Milligan and Cooper [44], which proved the superiority of the Rand index over Cohen's Kappa and also suggested a variety of validity metrics for internal consistency to examine the construct validity of the cluster solutions.

Only a part of the cited studies had a large representative sample of N > 2000 and none of the studies used more than one clustering algorithm. Moreover, with the exception of Herzberg and Roth [30] and Herzberg [33], none of the studies used a large variety of metrics for assessing internal and external consistency other than those provided by the respective clustering program they used. This limitation further adds up to the above mentioned bias towards smaller amounts of clusters although the field of cluster analysis and algorithms has developed a vast amount of internal and external validity algorithms and criteria to tackle this issue. Further, most of the studies had few or no other assessments or constructs than the Big Five to assess construct validity of the resulting personality types. Herzberg and Roth [30] and Herzberg [33] as well, though using a diverse variety of validity criteria only used one clustering algorithm on a medium-sized dataset with N < 2000.

Most of these limitations also apply to the study by Specht et. al. [36], which investigated two measurement occasions of the Big Five traits in the SOEP data sample. They used only one clustering algorithm (latent profile analysis), no other algorithmic validity criteria than the Bayesian information criterion and did not utilize any of the external constructs also assessed in the SOEP sample, such as mental health, locus of control or risk propensity for construct validation.

The largest sample and most advanced clustering algorithm was used in the recent study by Gerlach et al. [39]. But they also used only one clustering algorithm, and had no other variables except Big Five trait data to assess construct validity of the resulting personality types.

The aim of the present study was therefore to combine different methodological approaches while rectifying the shortcomings in several of the studies mentioned above in order to answer the following exploratory research questions: Are there replicable personality types, and if so, how many types are appropriate and in which constellations are they more (or less) useful than simple Big Five dimensions in the prediction of related constructs?

Three conceptually different clustering algorithms were used on a large representative dataset. The different solutions of the different clustering algorithms were compared using methodologically different internal and external validity criteria, in addition to those already used by the respective clustering algorithm.

To further examine the construct validity of the resulting personality types, their predictive validity in relation to physical and mental health, wellbeing, locus of control, self-esteem, impulsivity, risk-taking and patience were assessed.

Mental health and wellbeing seem to be associated mostly with neuroticism on the variable-oriented level [45], but on a person-oriented level, there seem to be large differences between the resilient and the overcontrolled personality type concerning perceived health and wellbeing beyond mean differences in neuroticism [33]. This seems also to be the case for locus of control and self-esteem, which is associated with neuroticism [46] and significantly differs between resilient and overcontrolled personality type [33]. On the other hand, impulsivity and risk taking seem to be associated with all five personality traits [47] and e.g. risky driving or sexual behavior seem to occur more often in the undercontrolled personality type [33, 48].

We chose these measures because of their empirically known differential associations to Big Five traits as well as to the above described personality types. So this both offers the opportunity to have an integrative comparison of the variable- and person-centered descriptions of personality and to assess construct validity of the personality types resulting from our analyses.

## Materials and methods

The acquisition of the data this study bases on was carried out in accordance with the principles of the Basel Declaration and recommendations of the "Principles of Ethical Research and Procedures for Dealing with Scientific Misconduct at DIW Berlin". The protocol was approved by the Deutsches Institut für Wirtschaftsforschung (DIW).

### Sample

The data used in this study were provided by the German Socio-Economic Panel Study (SOEP) of the German institute for economic research [49]. Sample characteristics are shown in Table 1. The overall sample size of the SOEP data used in this study, comprising all individuals who answered at least one of the Big-Five personality items in 2005 and 2009, was 25,821. Excluding all members with more than one missing answers on the Big Five assessment or intradimensional answer variance more than four times higher than the sample average resulted in a total Big Five sample of N = 22,820, which was used for the cluster analyses.

**Table 1. Sample characteristics.**

| Type | SOEP Wave | N | | Gender | Age (years) |
|---|---|---|---|---|---|
| Personality Type Derivation Sample | 2005, 2009 | Total | 25,821 | 51,8% F | M = 48.8, SD = 18.1, Rg = 17–100 |
| | | Included | 22,820 | 51,8% F | M = 48,6, SD = 18.0, Rg = 17–99 |
| Test Sample | 2013 | Total | 19,982 | 53.2% F | M = 52.3, SD = 17.9, Rg = 18–103 |
| | | Included | 17,549 | 53.2% F | M = 52.1, SD = 17.8, Rg = 18–103 |
| Longitudinal Construct Validity Sample | 2008, 2009, 2010 | Total | 14,048 | 52.0% F | M = 50.2, SD = 16.8, Rg = 20–98 |
| | | Included | 14,048 | | |

Exclusion of participants in the derivation and test samples based on missing answers or intradimensional answer variance more than four times higher than the sample average on the Big Five assessment. Longitudinal construct validity sample consistent of participants with available data on assessments of patience, risk taking, impulsivity, affective and cognitive wellbeing, locus of control, self-esteem and health. SOEP = German Socio-Economic Panel, M = mean, SD = standard deviation, Rg = Range, F = female.

14,048 of these individuals completed, in addition to the Big Five, items relevant to further constructs examined in this study that were assessed in other years. The 2013 SOEP data Big Five assessment was used as a test sample to examine stability and consistency of the final cluster solution.

## Measures

The Big Five were assessed in 2005 2009 and 2013 using the short version of the Big Five inventory (BFI-S). It consists of 15 items, with internal consistencies (Cronbach's alpha) of the scales ranging from .5 for openness to .73 for openness [50]. Further explorations showed strong robustness across different assessment methods [51].

To measure the predictive validity, several other measures assessed in the SOEP were included in the analyses. In detail, these were:

**Patience.** Patience was assessed in 2008 with one item: "Are you generally an impatient person, or someone who always shows great patience?"

**Risk taking.** Risk-taking propensity was assessed in 2009 by six items asking about the willingness to take risks while driving, in financial matters, in leisure and sports, in one's occupation (career), in trusting unknown people and the willingness to take health risks, using a scale from 0 (risk aversion) to 10 (fully prepared to take risks). Cronbach's alpha was .82 for this scale in the current sample.

**Impulsivity/Spontaneity.** Impulsivity/spontaneity was assessed in 2008 with one item: Do you generally think things over for a long time before acting–in other words, are you not impulsive at all? Or do you generally act without thinking things over for long time–in other words, are you very impulsive?

**Affective and cognitive wellbeing.** Affect was assessed in 2008 by four items asking about the amount of anxiety, anger, happiness or sadness experienced in the last four weeks on a scale from 1 (very rare) to 5 (very often). Cronbach's alpha for this scale was .66. The cognitive satisfaction with life was assessed by 10 items asking about satisfaction with work, health, sleep, income, leisure time, household income, household duties, family life, education and housing, with a Cronbach's alpha of .67. The distinction between cognitive and affective well-being stems from sociological research based on constructs by Schimmack et al. [50].

**Locus of control.** The individual attitude concerning the locus of control, the degree to which people believe in having control over the outcome of events in their lives opposed to being exposed to external forces beyond their control, was assessed in 2010 with 10 items, comprising four positively worded items such as "My life's course depends on me" and six

negatively worded items such as "Others make the crucial decisions in my life". Items were rated on a 7-point scale ranging from "does not apply" to "does apply". Cronbach's alpha in the present sample for locus of control was .57.

**Self-esteem.**  Global self-esteem–a person's overall evaluation or appraisal of his or her worth–was measured in 2010 with one item: "To what degree does the following statement apply to you personally?: I have a positive attitude toward myself".

**Health.**  To assess subjective health, the 12-Item Short Form Health Survey (SF-12) was integrated into the SOEP questionnaire and assessed in 2002, 2004, 2006, 2008 and 2010. In the present study, we used the data from 2008 and 2010. The SF-12 is a short form of the SF-36, a self-report questionnaire to assess the non-disease-specific health status [52]. Within the SF-12, items can be grouped onto two subscales, namely the physical component summary scale, with items asking about physical health correlates such as how exhausting it is to climb stairs, and the mental component summary scale, with items asking about mental health correlates such as feeling sad and blue. The literature on health measures often distinguishes between subjective and objective health measures (e.g., BMI, blood pressure). From this perspective, the SF-12 would count as a subjective health measure. In the present sample, Cronbach's alpha for the SF-12 items was .77.

## Derivation of the prototypes

The first step was to administer three different clustering methods on the Big Five data of the SOEP sample: First, the conventional linear clustering method used by Asendorpf [15, 35, 53] and also Herzberg and Roth [30] combines the hierarchical clustering method of Ward [54] with the k-means algorithm [55]. This algorithm generates a first guess of personality types based on hierarchical clustering, and then uses this first guess as starting points for the k-means-method, which iteratively adjusts the personality profiles, i.e. the cluster means to minimize the error of allocation, i.e. participants with Big Five profiles that are allocated to two or more personality types. The second algorithm we used was latent profile analysis with Mclust in R [56], an algorithm based on probabilistic finite mixture modeling, which assumes that there are latent classes/profiles/mixture components underlying the manifest observed variables. This algorithm generates personality profiles and iteratively calculates the probability of every participant in the data to be allocated to one of the personality types and tries to minimize an error term using maximum likelihood method. The third algorithm was spectral clustering, an algorithm which initially computes eigenvectors of graph Laplacians of the similarity graph constructed on the input data to discover the number of connected components in the graph, and then uses the k-means algorithm on the eigenvectors transposed in a k-dimensional space to compute the desired k clusters [57]. As it is an approach similar to the kernel k-means algorithm [58], spectral clustering can discover non-linearly separable cluster formations. Thus, this algorithm is able, in contrast to the standard k-means procedure, to discover personality types having unequal or non-linear distributions within the Big-Five traits, e.g. having a small SD on neuroticism while having a larger SD on conscientiousness or a personality type having high extraversion and either high or low agreeableness.

Within the last 50 years, a large variety of clustering algorithms have been established, and several attempts have been made to group them. In their book about cluster analysis, Bacher et al. [59] group cluster algorithms into incomplete clustering algorithms, e.g. Q-Sort or multi-dimensional scaling, deterministic clustering, e.g. k-means or nearest-neighbor algorithms, and probabilistic clustering, e.g. latent class and latent profile analysis. According to Jain [60], cluster algorithms can be grouped by their objective function, probabilistic generative models and heuristics. In his overview of the current landscape of clustering, he begins with the group

of density-based algorithms with linear similarity functions, e.g. DBSCAN, or probabilistic models of density functions, e.g. in the expectation-maximation (EM) algorithm. The EM algorithm itself also belongs to the large group of clustering algorithms with an information theoretic formulation. Another large group according to Jain is graph theoretic clustering, which includes several variants of spectral clustering. Despite the fact that it is now 50 years old, Jain states that k-means is still a good general-purpose algorithm that can provide reasonable clustering results.

The clustering algorithms chosen for the current study are therefore representatives of the deterministic vs. probabilistic grouping according to Bacher et. al. [59], as well as representatives of the density-based, information theoretic and graph theoretic grouping according to Jain [60].

## Determining the number of clusters

There are two principle ways to determine cluster validity: external or relative criteria and internal validity indices.

**External validity criteria.** External validity criteria measure the extent to which cluster labels match externally supplied class labels. If these external class labels originate from another clustering algorithm used on the same data sample, the resulting value of the external cluster validity index is relative. Another method, which is used in the majority of the cited papers in section 1, is to randomly split the data in two halves, apply a clustering algorithm on both halves, calculate the cluster means and allocate members of one half to the calculated clusters of the opposite half by choosing the cluster mean with the shortest Euclidean distance to the data member in charge. If the cluster algorithm allocation of one half is then compared with the shortest Euclidean distance allocation of the same half by means of an external cluster validity index, this results in a value for the reliability of the clustering method on the data sample.

As allocating data points/members by Euclidean distances always yields spherical and evenly shaped clusters, it will favor clustering methods that also yield spherical and evenly shaped clusters, as it is the case with standard k-means. The cluster solutions obtained with spectral clustering as well as latent profile analysis (LPA) are not (necessarily) spherical or evenly shaped; thus, allocating members of a dataset by their Euclidean distances to cluster means found by LPA or spectral clustering does not reliably represent the structure of the found cluster solution. This is apparent in Cohen's kappa values $<1$ if one uses the Euclidean external cluster assignment method comparing a spectral cluster solution with itself. Though by definition, Cohen's kappa should be 1 if the two ratings/assignments compared are identical, which is the case when comparing a cluster solution (assigning every data point to a cluster) with itself. This problem can be bypassed by allocating the members of the test dataset to the respective clusters by training a support vector machine classifier for each cluster. Support vector machines (SVM) are algorithms to construct non-linear "hyperplanes" to classify data given their class membership [61]. They can be used very well to categorize members of a dataset by an SVM-classifier trained on a different dataset. Following the rationale not to disadvantage LPA and spectral clustering in the calculation of the external validity, we used an SVM classifier to calculate the external validity criteria for all clustering algorithms in this study.

To account for the above mentioned bias to smaller numbers of clusters we applied three external validity criteria: Cohen's kappa, the Rand index [62] and the Hubert-Arabie adjusted Rand index [63].

**Internal validity criteria.** Again, to account for the bias to smaller numbers of clusters, we also applied multiple internal validity criteria selected in line with the the following

reasoning: According to Lam and Yan [64], the internal validity criteria fall into three classes: Class one includes cost-function-based indices, e.g. AIC or BIC [65], whereas class two comprises cluster-density-based indices, e.g. the S_Dbw index [66]. Class three is grounded on geometric assumptions concerning the ratio of the distances within clusters compared to the distances between the clusters. This class has the most members, which differ in their underlying mathematics. One way of assessing geometric cluster properties is to calculate the within- and/or between-group scatter, which both rely on summing up distances of the data points to their barycenters (cluster means). As already explained in the section on external criteria, calculating distances to cluster means will always favor spherical and evenly shaped cluster solutions without noise, i.e. personality types with equal and linear distributions on the Big Five trait dimensions, which one will rarely encounter with natural data.

Another way not solely relying on distances to barycenters or cluster means is to calculate directly with the ratio of distances of the data points within-cluster and between-cluster. According to Desgraupes [67], this applies to the following indices: the C-index, the Baker & Hubert Gamma index, the G(+) index, Dunn and Generalized Dunn indices, the McClain-Rao index, the Point-Biserial index and the Silhouette index. As the Gamma and G(+) indices rely on the same mathematical construct, one can declare them as redundant. According to Bezdek [68], the Dunn index is very sensitive to noise, even if there are only very few outliers in the data. Instead, the authors propose several ways to compute a Generalized Dunn index, some of which also rely on the calculation of barycenters. The best-performing GDI algorithm outlined by Bezdek and Pal [68] which does not make use of cluster barycenters is a ratio of the mean distance of every point between clusters to the maximum distance between points within the cluster, henceforth called GDI31. According to Vendramin et al. [69], the Gamma, C-, and Silhouette indices are the best-performing (over 80% correct hit rate), while the worst-performing are the Point-Biserial and the McClain-Rao indices (73% and 51% correct hit rate, respectively).

## Procedure

Fig 2 shows a schematic overview of the procedure we used to determine the personality types Big Five profiles, i.e. the cluster centers. To determine the best fitting cluster solution, we adopted the two-step procedure proposed by Blashfield and Aldenfelder [21] and subsequently used by Asendorpf [15, 35, 53] Boehm [41], Schnabel [24], Gramzow [28], and Herzberg and Roth [30], with a few adjustments concerning the clustering algorithms and the validity criteria.

First, we drew 20 random samples of the full sample comprising all individuals who answered the Big-Five personality items in 2005 and 2009 with N = 22,820 and split every sample randomly into two halves. Second, all three clustering algorithms described above were performed on each half, saving the 3-, 4-,. . .,9- and 10-cluster solution. Third, participants of each half were reclassified based on the clustering of the other half of the same sample, again for every clustering algorithm and for all cluster solutions from three to 10 clusters. In contrast to Asendorpf [35], this was implemented not by calculating Euclidean distances, but by training a support vector machine classifier for every cluster of a cluster solution of one half-sample and reclassifying the members of the other half of the same sample by the SVM classifier. The advantages of this method are explained in the section on external criteria. This resulted in 20 samples x 2 halves per sample x 8 cluster solutions x 3 clustering algorithms, equaling 960 clustering solutions to be compared.

The fourth step was to compute the external criteria comparing each Ward followed by k-means, spectral, or probabilistic clustering solution of each half-sample to the clustering by the

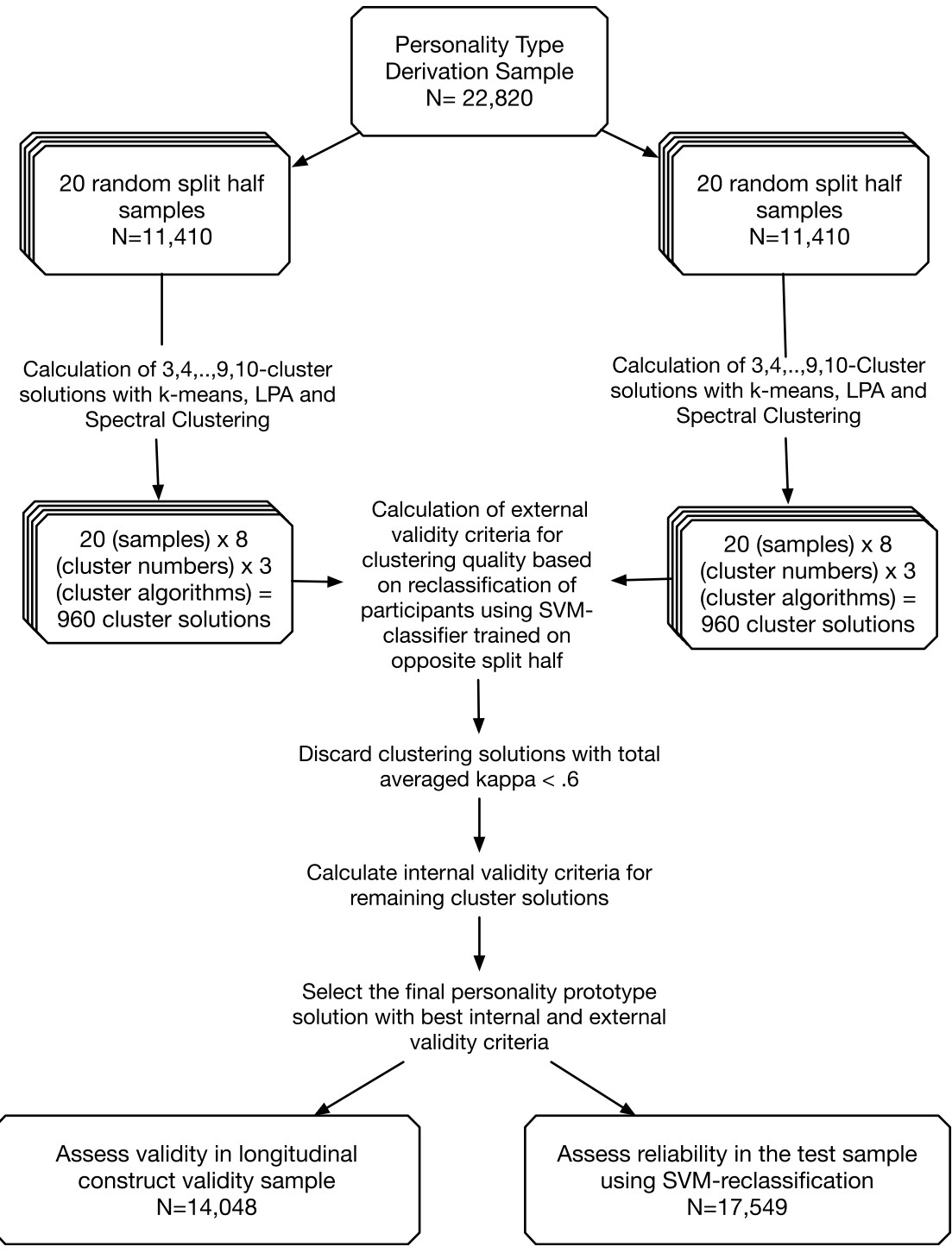

**Fig 2. Schematic overview of the procedure to determine the best fitting personality type solution.** LPA = latent profile analysis, SVM = Support Vector Machine.

SVM classifier trained on the opposite half of the same sample, respectively. The external cal-culated in this step were Cohen's kappa, Rand's index [62] and the Hubert & Arabie adjusted Rand index [63]. The fifth step consisted of averaging: We first averaged the external criteria

values per sample (one value for each half), and then averaged the 20x4 external criteria values for each of the 3-,4-..., 10-cluster solutions for each algorithm.

The sixth step was to temporarily average the external criteria values for the 3-,4-,... 10-cluster solution over the three clustering algorithms and discard the cluster solutions that had a total average kappa below 0.6.

As proposed by Herzberg and Roth [30], we then calculated several internal cluster validity indices for all remaining cluster solutions. The internal validity indices which we used were, in particular, the C-index [70], the Baker-Hubert Gamma index [71], the G + index [72], the Generalized Dunn index 31 [68], the Point-Biserial index [44], the Silhouette index [73], AIC and BIC [65] and the S_Dbw index [66]. Using all of these criteria, it is possible to determine the best clustering solution in a mathematical/algorithmic manner.

The resulting clusters where then assigned names by calculating Euclidean distances to the clusters/personality types found in the literature, taking the nearest type within the 5-dimensional space defined by the respective Big Five values.

To examine the stability and consistency of the final cluster solution, in a last step, we then used the 2013 SOEP data sample to calculate a cluster solution using the algorithm and parameters which generated the solution with the best validity criteria for the 2005 and 2009 SOEP data sample. The 2013 personality prototypes were allocated to the personality types of the solution from the previous steps by their profile similarity measure D. Stability then was assessed by calculation of Rand-index, adjusted Rand-index and Cohen's Kappa for the complete solution and for every single personality type. To generate the cluster allocations between the different cluster solutions, again we used SVM classifier as described above.

To assess the predictive and the construct validity of the resulting personality types, the inversed Euclidean distance for every participant to every personality prototype (averaged Big Five profile in one cluster) in the 5-dimensional Big-Five space was calculated and correlated with further personality, behavior and health measures mentioned above. To ensure that longitudinal reliability was assessed in this step, Big Five data assessed in 2005 were used to predict measures which where assessed three, four or five years later. The selection of participants with available data in 2005 and 2008 or later reduced the sample size in this step to N = 14,048.

## Results

### Internal and external cluster fit indices

Table 2 shows the mean Cohen's kappa values, averaged over all clustering algorithms and all 20 bootstrapped data permutations.

Whereas the LPA and spectral cluster solutions seem to have better kappa values for fewer clusters, the kappa values of the k-means clustering solutions have a peak at five clusters, which is even higher than the kappa values of the three-cluster solutions of the other two algorithms.

**Table 2. Averaged Cohen's kappa values for all cluster solutions with 3–10 clusters.**

|  | 3 Clusters | 4 Clusters | 5 Clusters | 6 Clusters | 7 Clusters | 8 Clusters | 9 Clusters | 10 Clusters |
|---|---|---|---|---|---|---|---|---|
| k-Means | 0.65 | 0.74 | 0.90 | 0.64 | 0.62 | 0.62 | 0.60 | 0.63 |
| LPA | 0.80 | 0.38 | 0.38 | 0.29 | 0.30 | 0.32 | 0.30 | 0.19 |
| Spectral | 0.89 | 0.88 | 0.81 | 0.70 | 0.76 | 0.75 | 0.75 | 0.67 |
| Mean | 0.78 | 0.66 | 0.70 | 0.54 | 0.56 | 0.56 | 0.55 | 0.50 |

Each cell is an average value over 20 independent cluster computations on random data permutations; the mean value in the last row is the average over all cluster algorithms. LPA = latent profile analysis, k-Means = k-Means Clustering algorithm, Spectral = Spectral clustering algorithm.

Considering that these values are averaged over 20 independent computations, there is very low possibility that this result is an artefact. As the solutions with more than five clusters had an average kappa below .60, they were discarded in the following calculations.

Table 3 shows the calculated external and internal validity indices for the three- to five-cluster solutions, ordered by the clustering algorithm. Comparing the validity criterion values within the clustering algorithms reveals a clear preference for the five-cluster solution in the spectral as well as the Ward followed by k-means algorithm.

Looking solely at the cluster validity results of the latent profile models, they seem to favor the three-cluster model. Yet, in a global comparison, only the S_Dbw index continues to favor the three-cluster LPA model, whereas the results of all other 12 validity indices support five-cluster solutions. The best clustering solution in terms of the most cluster validity index votes is the five-cluster Ward followed by k-means solution, and second best is the five-cluster spectral solution. It is particularly noteworthy that the five-cluster K-means solution has higher values on all external validity criteria than all other solutions. As these values are averaged over 20

**Table 3. Internal and external validity criterion values for the 3-, 4-, and 5-cluster solutions generated by the different clustering methods.**

| Validity Criterion | 3 Cluster Spectral | 4 Cluster Spectral | 5 Cluster Spectral | 3 Cluster LPA | 4 Cluster LPA | 5 Cluster LPA | 3 Cluster K-Means | 4 Cluster K-Means | 5 Cluster K-Means | Inter-pretation | Vote for |
|---|---|---|---|---|---|---|---|---|---|---|---|
| **Internal** | | | | | | | | | | | |
| C-Index | 0.266 | 0.242 | 0.220 | 0.176 | 0.161 | 0.153 | 0.128 | 0.113 | 0.102 | min | 5 Cluster K-Means |
| GDI31 | 0.336 | 0.320 | 0.338 | 0.145 | 0.137 | 0.135 | 0.173 | 0.178 | 0.176 | max | 5 Cluster Spectral |
| Baker-Hubert Gamma | 0.412 | 0.454 | 0.499 | 0.126 | 0.150 | 0.163 | 0.214 | 0.244 | 0.268 | max | 5 Cluster Spectral |
| G+ | 0.323 | 0.277 | 0.256 | 0.147 | 0.134 | 0.126 | 0.159 | 0.141 | 0.124 | min | 5 Cluster K-Means |
| Point Biserial | 0.388 | 0.386 | 0.399 | 0.120 | 0.132 | 0.137 | 0.199 | 0.206 | 0.208 | max | 5 Cluster Spectral |
| Silhouette | 0.178 | 0.165 | 0.174 | 0.026 | 0.021 | 0.031 | 0.084 | 0.079 | 0.080 | max | 3 Cluster Spectral |
| S_Dbw | 452606.0 | 803940.0 | 127385.0 | 159617.0 | 281694.0 | 425446.0 | 483406.0 | 708070.0 | 119891.0 | min | 3 Cluster LPA |
| AIC | 78059.8 | 72675.5 | 69713.5 | 41196.9 | 39966.8 | 39076.8 | 77718.8 | 73269.9 | 68945.2 | min | 5 Cluster LPA |
| BIC | 78300.8 | 72996.9 | 70115.3 | 41317.4 | 40127.5 | 39277.7 | 77959.9 | 73591.3 | 69346.9 | min | 5 Cluster LPA |
| **External** | | | | | | | | | | | |
| Cohen's Kappa | 0.892 | 0.877 | 0.808 | 0.796 | 0.378 | 0.383 | 0.649 | 0.737 | 0.898 | max | 5 Cluster K-Means |
| Rand Index | 0.909 | 0.913 | 0.892 | 0.841 | 0.691 | 0.723 | 0.821 | 0.863 | 0.939 | max | 5 Cluster K-Means |
| Hubert Arabie adjusted Rand | 0.818 | 0.827 | 0.784 | 0.683 | 0.381 | 0.447 | 0.642 | 0.725 | 0.877 | max | 5 Cluster K-Means |
| **Best value within algorithm count** | 3 | 2 | 7 | 5 | 0 | 7 | 2 | 1 | 9 | | 5 Clusters |
| **Best value between algorithm count** | 1 | 0 | 3 | 1 | 0 | 2 | 0 | 0 | 6 | | 5 Cluster K-Means |

Best value across all solutions for each validity criterion is highlighted in yellow, best value within the respective algorithm in blue. GDI31 = Generalized Dunn Index 31, AIC = Akaike's information criterion, BIC = Bayesian information criterion, LPA = latent profile analysis, k-Means = k-Means Clustering algorithm, Spectral = Spectral clustering algorithm.

independent cluster computations on random data permutations, and still have better values than solutions with fewer clusters despite the fact that these indices have a bias towards solutions with fewer clusters [42], there seems to be a substantial, replicable five-component structure in the Big Five Data of the German SOEP sample.

## Description of the prototypes

The mean z-scores on the Big Five factors of the five-cluster k-means as well as the spectral solution are depicted in Fig 2. Also depicted is the five-cluster LPA solution, which is, despite having poor internal and external validity values compared to the other two solutions, more complicated to interpret. To find the appropriate label for the cluster partitions, the respective mean z-scores on the Big Five factors were compared with the mean z-scores found in the literature, both visually and by the Euclidean distance.

The spectral and the Ward followed by k-means solution overlap by 81.3%; the LPA solution only overlaps with the other two solutions by 21% and 23%, respectively. As the Ward followed by k-means solution has the best values both for external and internal validity criteria, we will focus on this solution in the following.

The first cluster has low neuroticism and high values on all other scales and includes on average 14.4% of the participants (53.2% female; mean age 53.3, SD = 17.3). Although the similarity to the often replicated resilient personality type is already very clear merely by looking at the z-scores, a very strong congruence is also revealed by computing the Euclidean distance (0.61). The second cluster is mainly characterized by high neuroticism, low extraversion and low openness and includes on average 17.3% of the participants (54.4% female; mean age 57.6, SD = 18.2). It clearly resembles the overcontroller type, to which it also has the shortest Euclidean distance (0.58). The fourth cluster shows below-average values on the factors neuroticism, extraversion and openness, as opposed to above-average values on openness and conscientiousness. It includes on average 22.5% of the participants (45% female; mean age 56.8, SD = 17.6). Its mean z-scores closely resemble the reserved personality type, to which it has the smallest Euclidean distance (0.36). The third cluster is mainly characterized by low conscientiousness and low openness, although in the spectral clustering solution, it also has above-average extraversion and openness values. Computing the Euclidean distance (0.86) yields the closest proximity to the undercontroller personality type. This cluster includes on average 24.6% of the participants (41.3% female; mean age 50.8, *SD* = 18.3). The fifth cluster exhibits high z-scores on every Big Five trait, including a high value for neuroticism. Computing the Euclidean distances to the previously found types summed up in Fig 1 reveals the closest resemblance with the confident type (Euclidean distance = 0.81). Considering the average scores of the Big Five traits, it resembles the confident type from Herzberg and Roth [30] and Collani and Roth [10] as well as the resilient type, with the exception of the high neuroticism score. Having above average values on the more adaptive traits while having also above average neuroticism values reminded a reviewer from a previous version of this paper of the vulnerable but invincible children of the Kauai-study [74]. Despite having been exposed to several risk factors in their childhood, they were well adapted in their adulthood except for low coping efficiency in specific stressful situations. Taken together with the lower percentage of participants in the resilient cluster in this study, compared to previous studies, we decided to name the 5th cluster vulnerable-resilient. Consequently, only above or below average neuroticism values divided between resilient and vulnerable resilient. On average, 21.2% of the participants were allocated to this cluster (68.3% female; mean age 54.9, *SD* = 17.4).

Summarizing the descriptive statistics, undercontrollers were the "youngest" cluster whereas overcontrollers were the "oldest". The mean age differed significantly between clusters

($F$[4, 22820] = 116.485, $p$<0.001), although the effect size was small ($f$ = 0.14). The distribution of men and women between clusters differed significantly ($c^2$[4] = 880.556, $p$<0.001). With regard to sex differences, it was particularly notable that the vulnerable-resilient cluster comprised only 31.7% men. This might be explained by general sex differences on the Big Five scales. According to Schmitt et al. [75], compared to men, European women show a general bias to higher neuroticism (d = 0.5), higher conscientiousness (d = 0.3) and higher extraversion and openness (d = 0.2). As the vulnerable-resilient personality type is mainly characterized by high neuroticism and above-average z-scores on the other scales, it is therefore more likely to include women. In turn, this implies that men are more likely to have a personality profile characterized mainly by low conscientiousness and low openness, which is also supported by our findings, as only 41.3% of the undercontrollers were female.

Concerning the prototypicality of the five-cluster solution compared to the mean values extracted from previous studies, it is apparent that the resilient, the reserved and the overcontroller type are merely exact replications. In contrast to previous findings, the undercontrollers differed from the previous findings cited above in terms of average neuroticism, whereas the vulnerable-resilient type differed from the previously found type (labeled confident) in terms of high neuroticism.

## Stability and consistency

Inspecting the five cluster solution using the k-means algorithm on the Big Five data of the 2013 SOEP sample seemed to depict a replication of the above described personality types. This first impression was confirmed by the calculation of the profile similarity measure D between the 2005/2009 and 2013 SOEP sample cluster solutions, which yielded highest similarity for the undercontroler (D = 0.27) and reserved (D = 0.36) personality types, followed by the vulnerable-resilient (D = 0.37), overcontroler (D = 0.44) and resilient (D = 0.50) personality types. Substantial agreement was confirmed by the values of the Rand index (.84) and Cohen' Kappa (.70) whereas the Hubert Arabie adjusted Rand Index (.58) indicated moderate agreement for the comparison between the kmeans cluster solution for the 2013 SOEP sample and the cluster allocation with an SVM classifier trained on the 2005 and 2009 kmeans cluster solution.

## Predictive validity

In view of the aforementioned criticisms that (a) predicting dimensional variables will mathematically favor dimensional personality description models, and (b) using dichotomous predictors will necessarily provide less explanation of variance than a model using five continuous predictors, we used the profile similarity measure D [76] instead of dichotomous dummy variables accounting for the prototype membership. Correlations between the inversed Euclidean similarity measure D to the personality types and patience, risk-taking, spontaneity/impulsivity, locus of control, affective wellbeing, self-esteem and health are depicted in Table 4.

Patience had the highest association with the reserved personality type (r = .19, p < .001). The propensity to risky behavior, e.g. while driving (r = .17, p < .001), in financial matters (r = .17, p < .001) or in health decisions (r = .13, p < .001) was most highly correlated with the undercontroller personality type. This means that the more similar the Big-Five profile to the above-depicted undercontroller personality prototype, the higher the propensity for risky behavior. The average correlation across all three risk propensity scales with the undercontroller personality type is r = .21, with p < .001. This is in line with the postulations by Block and Block and subsequent replications by Caspi et al. [19, 48], Robins et al. [1] and Herzberg [33] about the undercontroller personality type. Spontaneity/impulsivity showed the highest

**Table 4. Pearson correlations between the inversed Euclidean distances of personality types constructed on the Big Five assessed in 2005 and patience, risk-taking, spontaneity/impulsivity, locus of control, self-esteem, wellbeing and health assessed longitudinally between 2008 and 2010.**

| | Inversed Euclidean Distance to Personality Type | | | | |
|---|---|---|---|---|---|
| Measure | Resilient | Vulnerable-Resilient | Undercontroller | Reserved | Overcontroller |
| Patience | .16 | -.04 | -.03 | **.19** | -**.07** |
| Risk-Taking (Mean) | .11 | (.01) | **.21** | -.03 | -**.06** |
| While Driving | .07 | (-.02) | **.17** | (-.02) | -.03 |
| Financial Matters | .07 | (.01) | **.17** | (.01) | (.01) |
| Health | (.01) | (.00) | **.13** | -.06 | (.00) |
| Spontaneity/ Impulsivity | **.15** | .07 | .12 | -.08 | -**.18** |
| (internal) Locus of Control | **.25** | -.06 | .03 | .10 | -**.22** |
| Self-Esteem | **.33** | (-.02) | .08 | .15 | -**.27** |
| Affective Wellbeing | **.27** | -.10 | .09 | .23 | -**.16** |
| Cognitive Wellbeing | **.28** | (-.02) | .07 | .13 | -**.21** |
| SF-12 Health (mean) | -**.23** | .08 | -.08 | -.16 | **.16** |
| Physical | -**.23** | .06 | -.10 | -.14 | **.16** |
| Mental | -.15 | .08 | (-.02) | -**.15** | **.11** |

N = 14048. Except the ones in brackets, only correlations with a significance level ≤ 0.001 are depicted. The highest and lowest correlation in each row are marked in bold. SF-12 = 12-Item Short Form Health Survey.

correlation with the overcontroller personality type (r = -.18, p<0.001). This is also in accordance with Block and Block, who described this type as being non-impulsive and appearing constrained and inhibited in actions and emotional expressivity.

Concerning locus of control, proximity to the resilient personality profile had the highest correlation with internal locus of control (r = .25, p < .001), and in contrast, the more similar the individual Big-Five profile was to the overcontroller personality type, the higher the propensity for external allocation of control (r = .22, p < .001). This is not only in line with Block and Block's postulations that the resilient personality type has a good repertoire of coping behavior and therefore perceives most situations as "manageable" as well as with the findings by [33], but is also in accordance with findings regarding the construct and development of resilience [77, 78].

Also in line with the predictions of Block and Block and replicating the findings of Herzberg [33], self-esteem was correlated the highest with the resilient personality profile similarity (r = .33, p < .001), second highest with the reserved personality profile proximity (r = .15, p < .001), and negatively correlated with the overcontroller personality type (r = -.27, p < .001).

This pattern also applies to affective and cognitive wellbeing as well as physical and mental health measured by the SF-12. Affective wellbeing was correlated the highest with similarity to the resilient personality type (r = .27, p < .001), and second highest with the reserved personality type (r = .23, p < .001). The overcontroller personality type, in contrast, showed a negative correlation with affective (r = -.16, p < .001) and cognitive (r = -21, p < .001) wellbeing. Concerning health, a remarkable finding is that lack of physical health impairment correlated the highest with the resilient personality profile similarity (p = -.23, p < .001) but lack of mental health impairment correlated the highest with the reserved personality type (r = -.15, p < .001). The highest correlation with mental health impairments (r = .11, p < .001), as well as physical health impairments (r = .16, p < .001) was with the overcontroller personality profile similarity. It is striking that although the undercontroller personality profile similarity was associated with risky health behavior, it had a negative association with health impairment measures, in contrast to the overcontroller personality type, which in turn had no association

with risky health behavior. This result is in line with the link of internalizing and externalizing behavior with the overcontroller and undercontroller types [79], respectively. Moreover, it is also in accordance with the association of internalizing problems with somatic symptoms and/ or symptoms of depressiveness and anxiety [80].

A further noteworthy finding is that these associations cannot be solely explained by the high neuroticism of the overcontroller personality type, as the vulnerable-resilient type showed a similar level of neuroticism but no correlation with self-esteem, the opposite correlation with impulsivity, and far lower correlations with health measures or locus of control. The vulnerable-resilient type showed also a remarkable distinction to the other types concerning the correlations to wellbeing. While for all other types, the direction and significance of the correlations to affective and cognitive measures of wellbeing were alike, the vulnerable-resilient type only had a significant negative correlation to affective wellbeing while having no significant correlation to measures of cognitive wellbeing.

To provide an overview of the particular associations of the Big Five values with all of the above-mentioned behavior and personality measures, Table 5 shows the bivariate correlations.

Investigating the direction of the correlation and the relativity of each value to each other row-wise reveals, to some extent, a clear resemblance with the z-scores of the personality types shown in Fig 3. Correlation profiles of risk taking, especially the facet risk-taking in health issues and locus of control, clearly resemble the undercontroller personality profile (negative correlations with openness and conscientiousness, positive but lower correlations with extraversion and openness). Patience had negative correlations with neuroticism and extraversion, and positive correlations with openness and conscientiousness, which in turn resembles the z-score profile of the reserved personality profile. Spontaneity/impulsivity had moderate to high positive correlations with extraversion and openness, and low negative correlations with openness and neuroticism, which resembles the inverse of the overcontroller personality profile. Self-esteem as well as affective and cognitive wellbeing correlations with the Big Five clearly resemble the resilient personality profile: negative correlations with neuroticism, and positive correlations with extraversion, openness, openness and conscientiousness. Inspecting the SF-12 health correlation, in terms of both physical and mental health, reveals a resemblance to the inversed resilient personality profile (high correlation with neuroticism, low correlation with

**Table 5. Pearson correlations between Big Five scores assessed in 2005 and patience, risk-taking, spontaneity/impulsivity, locus of control, self-esteem, wellbeing and health assessed longitudinally.**

| Measure | Neuroticism | Extraversion | Openness | Agreeablen. | Conscientousn. |
|---|---|---|---|---|---|
| Patience | -.23 | -.05 | (.00) | .28 | .08 |
| Risk-Taking (Mean) | -.11 | .11 | .17 | -.21 | -.12 |
| While Driving | -.10 | .10 | .09 | -.22 | -.08 |
| Financial Matters | -.09 | .03 | .10 | -.15 | -.11 |
| Health | -.03 | .04 | .10 | -.17 | -.16 |
| Spontaneity/ Impulsivity | -.04 | .27 | .18 | -.09 | .00 |
| (internal) Locus of Control | -.29 | .15 | .12 | .12 | .15 |
| Self-Esteem | -.33 | .22 | .15 | .16 | .18 |
| Affective Wellbeing | -.40 | .10 | .04 | .13 | .11 |
| Cognitive Wellbeing | -.28 | .15 | .13 | .12 | .16 |
| SF-12 Health (mean) | .32 | -.11 | -.06 | -.07 | -.11 |
| Physical | .30 | -.12 | -.10 | -.05 | -.08 |
| Mental | .25 | -.07 | .03 | -.07 | .13 |

N = 14,048. Except the one in brackets, only correlations with a significance level ≤ 0.001 are depicted. SF-12 = 12-Item Short Form Health Survey.

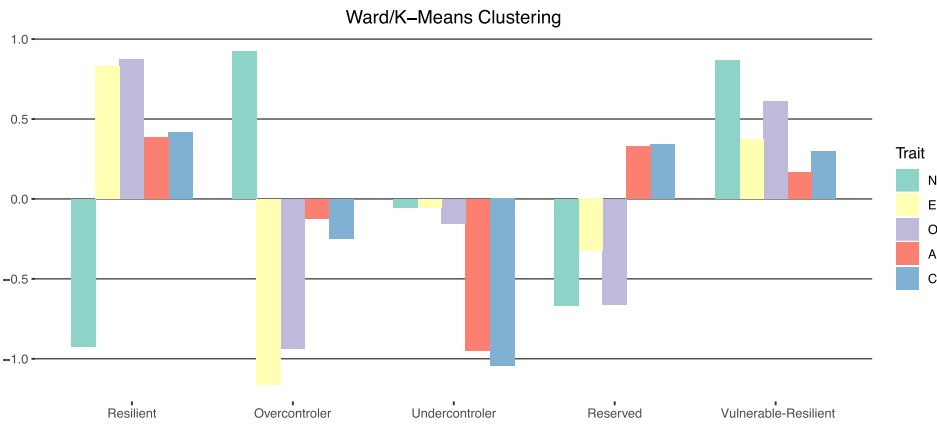

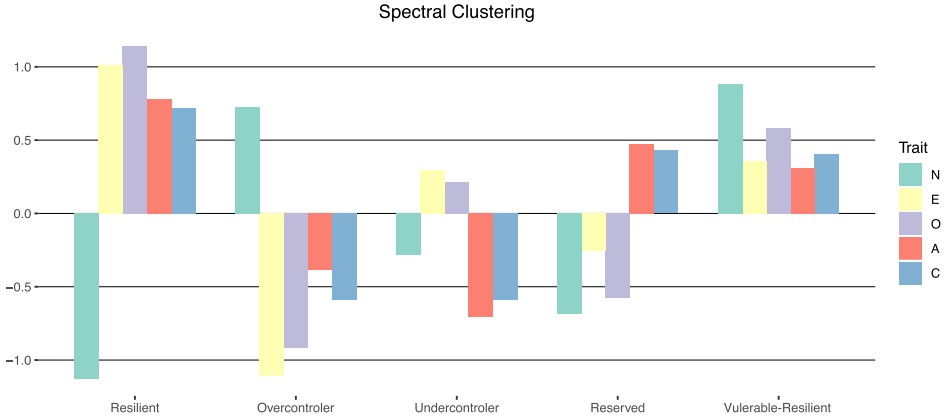

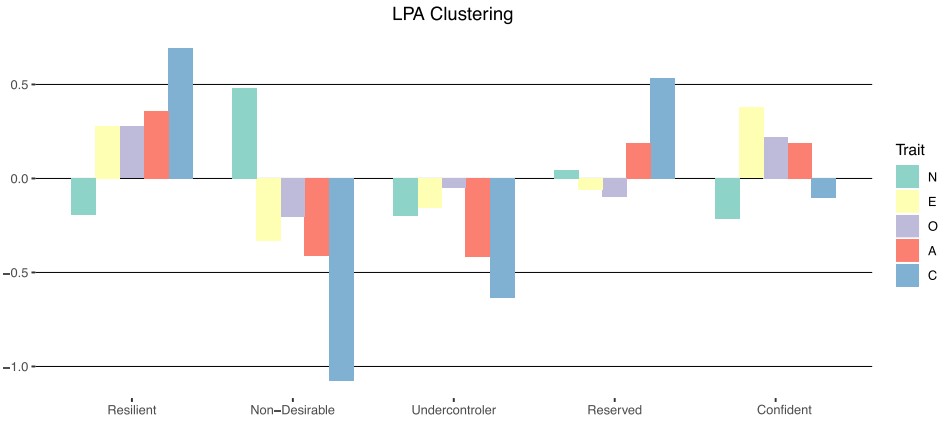

**Fig 3. Mean Big Five values of the five cluster solution calculated with the Ward followed by k- means, the spectral and latent profile analysis clustering algorithm.**

extraversion, openness, openness and conscientiousness, as well as a resemblance with the overcontroller profile (positive correlation with neuroticism, negative correlation with extraversion).

On the variable level, neuroticism had the highest associations with almost all of the predicted variables, with the exception of impulsivity, which was mainly correlated with extraversion and openness. It is also evident that all variables in question here are correlated with three or more Big Five traits. This can be seen as support for hypothesis that the concept of personality prototypes has greater utility than the variable-centered approach in understanding or predicting more complex psychological constructs that are linked to two or more Big Five traits.

## Discussion

The goal of this study was to combine different methodological approaches while overcoming the shortcomings of previous studies in order to answer the questions whether there are replicable personality types, how many of them there are, and how they relate to Big Five traits and other psychological and health-related constructs. The results revealed a robust five personality type model, which was able to significantly predict all of the psychological constructs in question longitudinally. Predictions from previous findings connecting the predicted variables to the particular Big Five dimensions underlying the personality type model were confirmed. Apparently, the person-centered approach to personality description has the most practical utility when predicting behavior or personality correlates that are connected to more than one or two of the Big Five traits such as self-esteem, locus of control and wellbeing.

This study fulfils all three criteria specified by von Eye & Bogat [81] regarding person-oriented research and considers the recommendations regarding sample size and composition by Herzberg and Roth [30]. The representative and large sample was analyzed under the assumption that it was drawn from more than one population (distinct personality types). Moreover, several external and internal cluster validity criteria were taken into account in order to validate the groupings generated by three different cluster algorithms, which were chosen to represent broad ranges of clustering techniques [60, 82]. The Ward followed by K-means procedure covers hierarchical as well as divisive partitioning (crisp) clustering, the latent profile algorithm covers density-based clustering with probabilistic models and information theoretic validation (AIC, BIC), and spectral clustering represents graph theoretic as well as kernel-based non-linear clustering techniques. The results showed a clear superiority of the five-cluster solution. Interpreting this grouping based on theory revealed a strong concordance with personality types found in previous studies, which we could ascertain both in absolute mean values and in the Euclidean distances to mean cluster z-scores extracted from 19 previous studies. As no previous study on personality types used that many external and internal cluster validity indices and different clustering algorithms on a large data set of this size, the present study provides substantial support for the personality type theory postulating the existence of resilient, undercontroller, overcontroller, vulnerable-resilient and reserved personality types, which we will refer to with RUO-VR subsequently. Further, our findings concerning lower validity of the LPA cluster solutions compared to the k-means and spectral cluster solutions suggest that clustering techniques based on latent models are less suited for the BFI-S data of the SOEP sample than iterative and deterministic methods based on the k-means procedure or non-linear kernel or graph-based methods. Consequently, the substance of the clustering results by Specht et. al. [36], which applied latent profile analysis on the SOEP sample, may therefore be limited.

But the question, if the better validity values of the k-means and spectral clustering techniques compared to the LPA indicate a general superiority of these algorithms, a superiority in the field of personality trait clustering or only a superiority in clustering this specific personality trait assessment (BFI-S) in this specific sample (SOEP), remains subject to further studies on personality trait clustering.

When determining the longitudinal predictive validity, the objections raised by Asendorpf [53] concerning the direct comparison of person-oriented vs. variable-oriented personality descriptions were incorporated by using continuous personality type profile similarity based on Cronbach and Gleser [75] instead of dichotomous dummy variables as well as by predicting long-term instead of cross-sectionally assessed variables. Using continuous profile similarity variables also resolves the problem that potentially important information about members of the same class is lost in categorical personality descriptions [15, 53, 83]. Predictions regarding the association of the personality types with the assessed personality and behavior correlates, including risk propensity, impulsivity, self-esteem, locus of control, patience, cognitive and affective wellbeing as well as health measures, were confirmed.

Overcontrollers showed associations with lower spontaneity/impulsivity, with lower mental and physical health, and lower cognitive as well as affective wellbeing. Undercontrollers were mainly associated with higher risk propensity and higher impulsive behavior. These results can be explained through the connection of internalizing and externalizing behavior with the overcontroller and undercontroller types [5–7, 78] and further with the connection of internalizing problems with somatic symptoms and/or symptoms of depressiveness and anxiety [79]. The dimensions or categories of internalizing and externalizing psychopathology have a long tradition in child psychopathology [84, 85] and have been subsequently replicated in adult psychopathology [86, 87] and are now basis of contemporary approaches to general psychopathology [88]. A central proceeding in this development is the integration of (maladaptive) personality traits into the taxonomy of general psychopathology. In the current approach, maladaptive personality traits are allocated to psychopathology spectra, such as the maladaptive trait domain negative affectivity to the spectrum of internalizing disorders. However, the findings of this study suggests that not specific personality traits are intertwined with the development or the occurrence of psychopathology but specific constellations of personality traits, in other words, personality profiles. This hypothesis is also supported by the findings of Meeus et al. [8], which investigated longitudinal transitions from one personality type to another with respect to symptoms of generalized anxiety disorder. Transitions from resilient to overcontroller personality profiles significantly predicted higher anxiety symptoms while the opposite was found for transitions from overcontroller to resilient personality profiles.

The resilient personality type had the strongest associations with external locus of control, higher patience, good health and positive wellbeing. This not only confirms the characteristics of the resilient type already described by Block & Block [18] and subsequently replicated, but also conveys the main characteristics of the construct of resilience itself. While the development of resiliency depends on the quality of attachment experiences in childhood and youth [89], resiliency in adulthood seems to be closely linked to internal locus of control, self-efficacy and self-esteem. In other words, the link between secure attachment experiences in childhood and resiliency in adulthood seems to be the development of a resilient personality trait profile. Seen the other way around, the link between traumatic attachment experiences or destructive environmental factors and low resiliency in adulthood may be, besides genetic risk factors, the development of personality disorders [90] or internalizing or externalizing psychopathology [91]. Following this thought, the p-factor [92], i.e. a general factor of psychopathology, may be an index of insufficient resilience. Although from the viewpoint of personality pathology, having a trait profile close to the resilient personality type may be an index of stable or good personality structure [93], i.e. personality functioning [94], which, though being consistently associated with general psychopathology and psychosocial functioning, should not be confused with it [95].

The reserved personality type had the strongest associations with higher patience as well as better mental health. The vulnerable-resilient personality type showed low positive

correlations with spontaneity/impulsivity and low negative correlations with patience as well as health and affective wellbeing.

Analyzing the correlations of the dimensional Big Five values with the predicted variables revealed patterns similar to the mean z-scores of the personality types resilient, overcontrollers, undercontrollers and reserved. Most variables had a low to moderate correlation with just one personality profile similarity, while having at least two or three low to moderate correlations with the Big Five measures. This can be seen as support for the argument of Chapman [82] and Asendorpf [15, 53] that personality types have more practical meaning in the prediction of more complex correlates of human behavior and personality such as mental and physical health, wellbeing, risk-taking, locus of control, self-esteem and impulsivity. Our findings further underline that the person-oritented approach may better be suited than variable-oriented personality descriptions to detect complex trait interactions [40]. E.g. the vulnerable-resilient and the overcontroller type did not differ in their high average neuroticism values, while differing in their correlations to mental and somatic health self-report measures. It seems that high neuroticism is far stronger associated to lower mental and physical health as well as wellbeing if it occurs together with low extraversion and low openness as seen in the overcontroller type. This differential association between the Big-Five traits also affects the correlation between neuroticism and self-esteem or locus of control. Not differing in their average neuroticism value, the overcontroller personality profile had moderate associations with low self-esteem and external locus of control while the vulnerable-resilient personality profile did only show very low or no association. Further remarkable is that the vulnerable-resilient profile similarity had no significant correlation with measures of cognitive wellbeing while being negatively correlated with affective wellbeing. This suggests that individuals with a Big-Five personality profile similar to the vulnerable-resilient prototype seem not to perceive impairments in their wellbeing, at least on a cognitive layer, although having high z-values in neuroticism. Another explanation for this discrepancy as well as for the lack of association of the vulnerable-resilient personality profile to low self-esteem and external locus of control though having high values in neuroticism could be found in the research on the construct of resilience. Personalities with high neuroticism values but stable self-esteem, internal locus of control and above average agreeableness and extraversion values may be the result of the interplay of multiple protective factors (e.g. close bond with primary caregiver, supportive teachers) with risk factors (e.g. parental mental illness, poverty). The development of a resilient personality profile with below average neuroticism values, on the other hand, may be facilitated if protective factors outweigh the risk factors by a higher ratio.

An interesting future research question therefore concerns to what extent personality types found in this study may be replicated using maladaptive trait assessments according to DSM-5, section III [96] or the ICD-11 personality disorder section [97] (for a comprehensive overview on that topic see e.g. [98]). As previous studies showed that both DSM-5 [99] and ICD-11 [100] maladaptive personality trait domains may be, to a large extent, conceptualized as maladaptive variants of Big Five traits, it is highly likely that also maladaptive personality trait domains align around personality prototypes and that the person-oriented approach may amend the research field of personality pathology [101].

Taken together, the findings of this study connect the variable centered approach of personality description, more precisely the Big Five traits, through the concept of personality types to constructs of developmental psychology (resiliency, internalizing and externalizing behavior and/or problems) as well as clinical psychology (mental health) and general health assessed by the SF-12. We could show that the distribution of Big Five personality profiles, at least in the large representative German sample of this study, aggregates around five prototypes, which in

turn have distinct associations to other psychological constructs, most prominently resilience, internalizing and externalizing behavior, subjective health, patience and wellbeing.

## Limitations

Several limitations of the present study need to be considered: One problem concerns the assessment of patience, self-esteem and impulsivity. From a methodological perspective, these are not suitable for the assessment of construct validity as they were assessed with only one item. A further weakness is the short Big Five inventory with just 15 items. Though showing acceptable reliability, 15 items are more prone to measurement errors than measures with more items and only allow a very broad assessment of the 5 trait domains, without information on individual facet expressions. A more big picture question is if the Big Five model is the best way to assess personality in the first place. A further limitation concerns the interpretation of the subjective health measures, as high neuroticism is known to bias subjective health ratings. But the fact that the vulnerable-resilient and the overcontroler type had similar average neuroticism values but different associations with the subjective health measures speaks against a solely neuroticism-based bias driven interpretation of the associations of the self-reported health measures with the found personality clusters. Another limitation is the correlation between the personality type similarities: As they are based on Euclidean distances and the cluster algorithms try to maximize the distances between the cluster centers, proximity to one personality type (that is the cluster mean) logically implies distance from the others. In the case of the vulnerable-resilient and the resilient type, the correlation of the profile similarities is positive, as they mainly differ on only one dimension (neuroticism). These high correlations between the profile similarities prevents or diminishes, due to the emerging high collinearity, the applicability of general linear models, i.e. regression to calculate the exact amount of variance explained by the profile similarities.

The latter issue could be bypassed by assessing types and dimensions with different questionnaires, i.e. as in Asendorpf [15] with the California Child Q-set to determine the personality type and the NEO-FFI for the Big Five dimensions. Another possibility is to design a new questionnaire based on the various psychological constructs that are distinctly associated with each personality type, which is probably a subject for future person-centered research.

## Acknowledgments

The data used in this article were made available by the German Socio-Economic Panel (SOEP, Data for years 1984–2015) at the German Institute for Economic Research, Berlin, Germany. However, the findings and views reported in this article are those of the authors. To ensure the confidentiality of respondents' information, the SOEP adheres to strict security standards in the provision of SOEP data. The data are reserved exclusively for research use, that is, they are provided only to the scientific community. All users, both within the EEA (and Switzerland) and outside these countries, are required to sign a data distribution contract.

## Author Contributions

**Conceptualization:** André Kerber, Marcus Roth, Philipp Yorck Herzberg.

**Data curation:** André Kerber.

**Formal analysis:** André Kerber.

**Investigation:** André Kerber, Marcus Roth, Philipp Yorck Herzberg.

**Methodology:** André Kerber, Marcus Roth, Philipp Yorck Herzberg.

**Supervision:** Marcus Roth, Philipp Yorck Herzberg.

**Validation:** André Kerber, Marcus Roth, Philipp Yorck Herzberg.

**Visualization:** André Kerber.

**Writing – original draft:** André Kerber.

**Writing – review & editing:** Marcus Roth, Philipp Yorck Herzberg.

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
