## [Decision Letter · Decision Letter 0]

24 Mar 2020

PONE-D-20-00337

Personality Types Revisited – a Comprehensive Algorithmic Approach to an Integration of Prototypical and Dimensional Constructs of Personality Description

PLOS ONE

Dear Kerber,

Thank you for submitting your manuscript to PLOS ONE. After careful consideration, we feel that it has merit but does not fully meet PLOS ONE’s publication criteria as it currently stands. Therefore, we invite you to submit a revised version of the manuscript that addresses the points raised during the review process.

We would appreciate receiving your revised manuscript by April 23, 2020. To enhance the reproducibility of your results, we recommend that if applicable you deposit your laboratory protocols in protocols.io, where a protocol can be assigned its own identifier (DOI) such that it can be cited independently in the future. For instructions see: http://journals.plos.org/plosone/s/submission-guidelines#loc-laboratory-protocols

We look forward to receiving your revised manuscript.

Kind regards,

Stephan Doering, M.D.

Academic Editor

PLOS ONE

Reviewers' comments:

Reviewer's Responses to Questions

**Comments to the Author**

1. Is the manuscript technically sound, and do the data support the conclusions?

Reviewer #1: Yes

Reviewer #2: Yes

2. Has the statistical analysis been performed appropriately and rigorously? 

Reviewer #1: Yes

Reviewer #2: Yes

3. Have the authors made all data underlying the findings in their manuscript fully available?

Reviewer #1: No

Reviewer #2: Yes

4. Is the manuscript presented in an intelligible fashion and written in standard English?

Reviewer #1: Yes

Reviewer #2: Yes

5. Review Comments to the Author

Reviewer #1: This is a sophisticated and methodologically exhaustive study.

I must confess that much of this material is beyond my scope of knowledge and my capacity to comprehend.

After reading the abstract I am still not very clear about what this study is all about. It all seems like at novel approach - or maybe a pioneering approach is a more appropriate term. Therefore, I suggest that nothing should be too obvious in the communication of this study.

1] For example, the abstract only refers to “a large representative German dataset” without providing the N? What is the abbreviation Ward/k standing for?

In general, I encourage the authors to rephrase the abstract and parts of the introduction as a service for the reader.

2] General comment: During the introduction on the first 3-4 pages, I feel somewhat lost as reader. I suppose the authors could sharpen up this part. It may also be helpful to link the rationale to some more familiar/contemporary theory and research within the field.

The 10 first pages seem to work as a review of the literature.

The aim is not presented until page 9 line 207.

3] It could be helpful with a more clear distinguishing between types and traits?

4] Page 6, line 127: “In a recent nature human behavior publication” – are the authors referring to a journal here or a particular issue or paper? It is not evident.

5] Page 7: ”total N of 1560418” – please use comma separators.

6] The authors cite the HiTOP and related scientific papers (e.g., Forbush et al, Kotov et al., Krueger et al.). However, the authors did not relate their findings or discussions to the more authoritative diagnostic frameworks such as the approved ICD-11 dimensional classifications of PDs as well as the DSM-5 alternative model – with particular emphasis on their trait systems.

7] On page 4 the authors write: “it can be said that the human goal is to be as undercontrolled as possible and as overcontrolled as necessary. When one is more undercontrolled than is adaptively effective or more overcontrolled than is adaptively required, one is not resilient”

In relation to “resilience”, it is remarkable that the authors have not related their findings or discussion to Fonagy and Sharp as well as Caspi’s P-factor (see references below). I particularly refer to the P-factor as an index of insufficient resilience, which may be something that could be more clearly incorporated into the manuscript?

Caspi, A., Houts, R. M., Belsky, D. W., & Goldman-mellor, S. J. (2015). The p factor: One general psychopathology factor in the structure of psychiatric disorders? Clinical Psychological Science, 2(2), 119–137. https://doi.org/10.1177/2167702613497473.The

Sharp, C., Wright, A. G. C., Fowler, J. C., Frueh, B. C., Allen, J. G., Oldham, J., & Clark, L. A. (2015). The structure of personality pathology: Both general (‘g’) and specific (‘s’) factors? Journal of Abnormal Psychology, 124(2), 387–398. https://doi.org/10.1037/abn0000033

Fonagy, P., Luyten, P., Allison, E., & Campbell, C. (2017). What we have changed our minds about: Part 1. Borderline personality disorder as a limitation of resilience. Borderline Personality Disorder and Emotion Dysregulation, 4(1), 11. https://doi.org/10.1186/s40479-017-0061-9

Reviewer #2: congratulations to the authors, this is an excellent work which, however, has two fundamental limitations: 1. it includes a long part, not consistent with the title and the abstract, which can be eliminated; 2 the description of the statistical methodology is poorly understood by colleagues who are not experts in data analysis. The text is weighted and complex to read.

I will point out my thoughts step by step. following them the writing becoming more agile and accessible will bring out the fantastic work behind it.

from row 48 to row 51

The difference between the two approaches should be clearly explained

from row 69 to row 70

The Q procedure should be clearly explained

from row 88 to row 91

I would delete this sentence

from row 127 to row 130

I would explain this study in more detail

rom row 131 to row 133

the reasons for this choice should be explained

from row 143 to row 206

I would eliminate this part

(it seems to me, to all intents and purposes, something that may belong to an interesting review of the literature. this part proposed in this stringed way is obviously inadequate, inconsistent with the title and unnecessarily burdens the text)

Clearly this implies the elimination also of figure 1 and of the results and discussion that refer to the comparison between figure 1 and figure 2

from row 227 to row 229

I would extend this concept

from row 233 to row 244

I would insert a table representing the elements described

from row 291 to row 321

from row 325 to row 353

The meaning of these methods should be clarified in relation to the type of data examined. this will allow a perfect understanding of the results even for non-expert colleagues in data analysiss

from row 354 to row 357

I would delete this sentence, there are recent studies that question this claim

(see Matthijs J Warrens On the Equivalence of Cohen’s Kappa and the Hubert-Arabie Adjusted Rand Index

February 2008 Journal of Classification 25 (2): 177-183)

from row 358 to row 383

These are basic concepts for which the paragraph can be reduced in size

from row 384 to row 435

Insert a figure that graphically describes the procedure

rom row 677 to row 680

delete this sentence

6. PLOS authors have the option to publish the peer review history of their article (what does this mean?). If published, this will include your full peer review and any attached files.

Reviewer #1: No

Reviewer #2: Yes: Raffaele Sperandeo

---

## [Author Response · Author response to Decision Letter 0]

23 Sep 2020

Please see the attached document "Response to the reviewers".

---

## [Decision Letter · Decision Letter 1]

9 Nov 2020

PONE-D-20-00337R1

Personality Types Revisited – a Literature-Informed and Data-Driven Approach to an Integration of Prototypical and Dimensional Constructs of Personality Description

PLOS ONE

Dear Dr. Kerber,

Thank you for submitting your manuscript to PLOS ONE. After careful consideration, we feel that it has merit but does not fully meet PLOS ONE’s publication criteria as it currently stands. Therefore, we invite you to submit a revised version of the manuscript that addresses the points raised during the review process. There are only very minor points raised by reviewer 1 that need to be addressed.

We look forward to receiving your revised manuscript.

Kind regards,

Stephan Doering, M.D.

Academic Editor

PLOS ONE

Reviewers' comments:

Reviewer's Responses to Questions

**Comments to the Author**

1. If the authors have adequately addressed your comments raised in a previous round of review and you feel that this manuscript is now acceptable for publication, you may indicate that here to bypass the “Comments to the Author” section, enter your conflict of interest statement in the “Confidential to Editor” section, and submit your "Accept" recommendation.

Reviewer #1: All comments have been addressed

Reviewer #2: All comments have been addressed

2. Is the manuscript technically sound, and do the data support the conclusions?

Reviewer #1: Yes

Reviewer #2: Yes

3. Has the statistical analysis been performed appropriately and rigorously? 

Reviewer #1: Yes

Reviewer #2: Yes

4. Have the authors made all data underlying the findings in their manuscript fully available?

Reviewer #1: No

Reviewer #2: Yes

5. Is the manuscript presented in an intelligible fashion and written in standard English?

Reviewer #1: Yes

Reviewer #2: Yes

6. Review Comments to the Author

Reviewer #1: I feel the authors overall adressed the issues I raised.

I only have the following minor comments:

1) The tables have no definitions in the legend for the different terms and abbreivations - I am not entirely aware of the author guidelines for this journal, but I think it is much needed.

2) The authors rigthly included a reference to the now approved ICD-11 PD classification (line 825). However, the authors should provide the correct reference:

WHO. (2019). ICD-11 Clinical Descriptions and Diagnostic Guidelines for Mental and Behavioural Disorders. World Health Organisation. gcp.network/en/private/icd-11-guidelines/disorders

3) Moreover, they only refer to studies on big five convergence with DSM-5 Section III traits - but not with the ICD-11 traits. See for example the following papers:

Somma, A., Gialdi, G., & Fossati, A. (2020). Reliability and construct validity of the Personality Inventory for ICD-11 (PiCD) in Italian adult participants. Psychological Assessment, 32(1), 29–39. https://doi.org/10.1037/pas0000766

Oltmanns, J. R., & Widiger, T. A. (2018). A self-report measure for the ICD-11 dimensional trait model proposal: The Personality Inventory for ICD-11. Psychological Assessment, 30(2), 154–169. https://doi.org/10.1037/pas0000459

Oltmanns, J. R., & Widiger, T. A. (2019). The Five-Factor Personality Inventory for ICD-11: A facet-level assessment of the ICD-11 trait model. Psychological Assessment. https://doi.org/10.1037/pas0000763

Reviewer #2: I read this study and reviewed it with great pleasure. I congratulate you on this innovative work which appears to be a milestone in the study of personality

7. PLOS authors have the option to publish the peer review history of their article (what does this mean?). If published, this will include your full peer review and any attached files.

Reviewer #1: No

Reviewer #2: No

---

## [Author Response · Author response to Decision Letter 1]

13 Dec 2020

Reviewer 1:

1) The tables have no definitions in the legend for the different terms and abbreivations - I am not entirely aware of the author guidelines for this journal, but I think it is much needed.

Response: Thanks to this suggestion we have reviewed all our tables for abbreviations that are not explained and included them in the respective notes.

2) The authors rigthly included a reference to the now approved ICD-11 PD classification (line 825). However, the authors should provide the correct reference:

Response: We have rephrased ll. 838-841 to also include a reference to the ICD-11 PD model.

---

## [Editor Report · Decision Letter 2]

18 Dec 2020

Personality Types Revisited – a Literature-Informed and Data-Driven Approach to an Integration of Prototypical and Dimensional Constructs of Personality Description

PONE-D-20-00337R2

Dear Dr. Kerber,

We’re pleased to inform you that your manuscript has been judged scientifically suitable for publication and will be formally accepted for publication once it meets all outstanding technical requirements.

Kind regards,

Stephan Doering, M.D.

Academic Editor

PLOS ONE

---

## [Editor Report · Acceptance letter]

22 Dec 2020

PONE-D-20-00337R2 

Personality Types Revisited – a  Literature-Informed and Data-Driven Approach to an Integration of Prototypical and Dimensional Constructs of Personality Description 

Dear Dr. Kerber:

I'm pleased to inform you that your manuscript has been deemed suitable for publication in PLOS ONE. Congratulations! Your manuscript is now with our production department. 

Kind regards, 

on behalf of

Professor Stephan Doering 

Academic Editor

PLOS ONE